# Cogging Torque Reduction Based on a New Pre-Slot Technique for a Small Wind Generator

**Miguel García-Gracia** [1,*] **, Ángel Jiménez Romero** [1] **, Jorge Herrero Ciudad** [2] **and Susana Martín Arroyo** [1]

[1] Department of Electrical Engineering, University of Zaragoza, 50018 Zaragoza, Spain; anjirom@unizar.es (Á.J.R.); smartin@unizar.es (S.M.A.)

[2] For Optimal Renewable Energy Systems, S.L. (4fores), 50197 Zaragoza, Spain; herrero@4fores.es

[*] Correspondence: mggracia@unizar.es; Tel.: +34-976-761923

**Abstract:** Cogging torque is a pulsating, parasitic, and undesired torque ripple intrinsic of the design of a permanent magnet synchronous generator (PMSG), which should be minimized due to its adverse effects: vibration and noise. In addition, as aerodynamic power is low during start-up at low wind speeds in small wind energy systems, the cogging torque must be as low as possible to achieve a low cut-in speed. A novel mitigation technique using compound pre-slotting, based on a combination of magnetic and non-magnetic materials, is investigated. The finite element technique is used to calculate the cogging torque of a real PMSG design for a small wind turbine, with and without using compound pre-slotting. The results show that cogging torque can be reduced by a factor of 48% with this technique, while avoiding the main drawback of the conventional closed slot technique: the reduction of induced voltage due to leakage flux between stator teeth. Furthermore, through a combination of pre-slotting and other cogging torque optimization techniques, cogging torque can be reduced by 84% for a given design.

**Keywords:** cogging torque; permanent magnet synchronous generator; small wind turbines; finite element method; renewable energy; energy conversion

## 1. Introduction

Increasing interest in the efficiency of electric machinery and reducing maintenance costs is making the use of permanent magnet synchronous generators (PMSGs) more common. PMSGs combine high efficiency with low maintenance and a high power density [1], factors that make them extremely attractive for use in renewable energy applications are, such as wind [2], wave power [3], and tidal power [4], or electrical mobility applications [5] and, in general, in uses where they must act as a motor or generator. Furthermore, in renewable energy applications, PMSGs allow direct-drive configurations, making the use of a gearbox unnecessary or reducing the number of gearbox stages, which decreases the overall generator volume and improves its efficiency [6].

However, machines based on permanent magnets (PMs) also have some drawbacks, and the cogging torque is one of the main ones. The magnetic interaction between the flux generated by the rotor PMs and the stator geometry results in a pulsating torque called cogging torque, which, depending on the PM machine design, can cause an undesired ripple in both the machine's induced voltage (EMF) and its mechanical torque [7,8]. Other problems with PMSGs are the vibrations and noise they make. Since this type of machine has high magnetic flux density values in the air gap, the electromagnetic forces between the PMs and the stator teeth are high [9]. These electromagnetic forces are divided into two components, one radial and the other tangential. The tangential component of the electromagnetic force contributes to the torque in the stator teeth, while the radial component

causes vibrations and even deformations in the machine [10]. These radial forces act on the stator producing vibrations and noise, especially when their frequency coincides with the natural frequency of the machine's mechanical structure [11].

The cogging torque is especially important in wind energy applications as it establishes in which conditions the system will begin generating. The mechanical torque captured by the generation system must be larger than the cogging torque starting the rotation, which is why achieving a reduced cogging torque is one of the objectives for this type of machine.

There are several methods to reduce cogging torque in the PMSG design phase. The most used is skewing, which consists of preventing the stator teeth and the magnets from becoming aligned by either turning the stator teeth [6,12] or the rotor's permanent magnets [1,2,13]. The required skew angle to largely cancel out the effect of the interactions between the PMs and the slots depends on how many slots and poles the machine has. Other methods study the use of notches in stator teeth [7,14]. These notches produce the same effect in the magnetic interaction as the slots and increase the effective number of slots, which impacts on the cogging torque as it depends on how many poles and slots the machine has. Therefore, this method's effectiveness is conditioned by the number of poles and slots selected in the design.

A study is presented in [15,16] in which slot openings in one half of the stator, shift in one direction with respect to the tooth and in the other half, shift in the other direction. This means the cogging torque waveform moves in opposite directions in each machine half and the cogging caused by each machine half may be cancelled out depending on the shifted angle. Other studies focus on the shape of PM edges, concluding that their size can be reduced on the magnet sides to lessen air gap reluctance variation, which reduces the magnetic energy variation in the machine and, therefore, mitigates the cogging torque [17,18].

Several authors have conducted studies of PMSG with closed slots and their effects. Leakage fluxes caused as a result of closing stator slots are analyzed in [4], concluding that the size of PMs should be increased to compensate flux loss through closed slots. The increase in iron losses caused by tooth-tip saturation, distortion in the induced voltage this saturation causes and how the use of closed slots influences this, are studied in [19,20]. The study by [21] focuses on average torque and its ripple in machines with closed slots for several stator types. The work by [22] analyzes the use of magnetic composite wedges to close stator slots in terms of stator flux linkage, average and ripple torque, and magnetic losses.

Unlike the above-mentioned methods, which focus on minimizing the cogging torque in the machine design stage, this article proposes a cogging torque reduction method that is easy to implement without the need for any changes to the original design of the machine, a 6.3-kW generator for a small wind turbine. The suggested solution comprises sliding a metal part (a pre-slot wedge) into the slots after completing the machine winding. This technique minimizes slot openings so that induced voltage remains unaltered and the mounting of machine windings is not hampered. The results of the proposed method are analyzed using FEMM 2D (Finite Element Method Magnetics) software on an original PMSG design and compared with the results obtained experimentally. Additionally, constructive improvements are suggested to reduce cogging torque. Finally, the article shows how the proposed technique can also be combined with the skewing technique, thus significantly reducing the cogging torque to 0.03 Nm in the ideal case and 0.51 Nm when imperfections in the manufacturing process are considered.

## 2. Machine Type and Main Parameters

The machine involved in this study is a double layer fractional winding PMSG with an interior rotor and surface-mounted magnets comprising 36 slots and 20 poles. Figure 1a is a cross section of the generator showing the slots forming the stator, the rotor in the internal part, and the surface-mounted magnets above it in the central part. The details of the millimeter measurements of the machine's

slots and PMs are shown in Figure 2. The characteristic parameters of the studied PMSG are shown in Table 1.

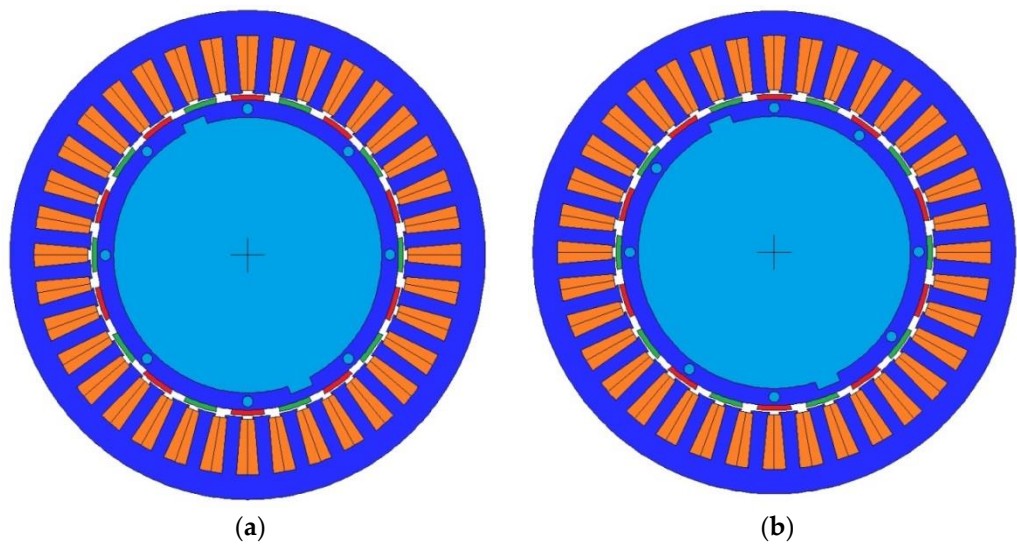

**Figure 1.** Cross section area of the permanent magnet synchronous generator (PMSG): (**a**) Original model; (**b**) Model with centered holes.

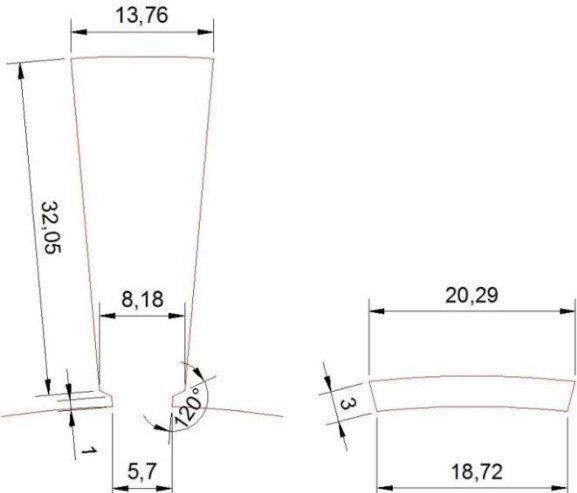

**Figure 2.** Slot and magnet dimensions (mm).

**Table 1.** Parameters of the permanent magnet synchronous generator (PMSG) machine.

| Parameter | Value |
|---|---|
| Phase | 3 |
| Pole number | 20 |
| Slot number | 36 |
| Rated speed | 232 rpm |
| Rated power | 6300 W |
| Rated voltage | 256.4 V |
| Rated torque | 102 Nm |
| Air gap | 1 mm |
| Thickness of PM | 3 mm |
| Rotor diameter | 180 mm |
| Material of steel | M330-50A |
| Material of PM | NdFeB |

The analysis of the PMSG and the cogging torque reduction methods proposed in this study was performed using FEMM 2D finite element software. To validate the FEMM model used in the cogging torque reduction analysis a comparison is made with the experimental values of the original machine.

## 3. Cogging Torque

Cogging torque is a parasitic torque resulting from interactions between the rotor's permanent magnets and the stator slots. Air gap reluctance differs depending on the rotor's angular position to the slots. Rotor magnets tend to align with the stator in the position in which air gap permeance is larger [23], so when they are shifted from this position during rotation, they generate a torque, the cogging torque.

Electromagnetic torque can be obtained from the variation in the total energy of the magnetic field compared with the angular position of the rotor $\theta$ when excitation current is constant [14]

$$T = -\frac{\partial W_c}{\partial \theta} \tag{1}$$

The total energy stored in the magnetic field or coenergy $W_c$ in a PMSG is given by [7]

$$W_c = \frac{1}{2} L i^2 + \frac{1}{2}(R + R_m) \varnothing_m^2 + N i \varnothing_m \tag{2}$$

where $L$ is the inductance of the windings, $i$ the excitation current, $R$ and $R_m$ are, respectively, the reluctances viewed by the magnetomotive force and by the magnetic field, $\varnothing_m$ the flux due to the magnets crossing the air gap, and $N$ the number of winding turns.

Therefore, substituting in Equation (2) results in

$$T = \frac{1}{2} i^2 \frac{dL}{d\theta} - \frac{1}{2} \varnothing_m^2 \frac{dR}{d\theta} + N i \frac{d\varnothing_m}{d\theta} \tag{3}$$

The second term of Equation (3) corresponds to magnet reluctance torque and it is known as cogging torque [17], $T_{cog}$

$$T_{cog} = -\frac{1}{2} \varnothing_m{}^2 \frac{dR}{d\theta} \tag{4}$$

As observed in Equation (4), cogging torque is independent of the current and corresponds to the result of analyzing Equation (3). when the machine is in open circuit. Cogging torque depends on magnetic flux and on the rate of change of air-gap reluctance. From Equation (4), to minimize $T_{cog}$, reluctance $R$ should be independent of the rotor position. Therefore, a very low cogging torque design requires an almost constant value of $R$ for any rotor position.

## 4. Cogging Torque Measurement

Cogging torque is calculated in FEMM for every angular rotor position, making the machine operate off-load. The torque is calculated by integrating the Maxwell stress tensor throughout the air gap

$$T_{cog} = \frac{L}{g\, u_o} \int\limits_S r\, B_n\, B_t\, dS \tag{5}$$

where $\mu_o$ is the air gap permeability, $L$ is the rotor depth, $g$ is the air gap length, $B_n$ the normal flux density, $B_t$ the tangential flux density and $r$ the radius from the center of the rotor to the center of the air gap [7].

Compared with the results obtained when the calculation is based on the magnetic energy variation with respect to the angular rotor position given by Equation (1), in [18] it is shown that both methods obtain almost identical results.

The simulation in FEMM of the PMSG in Figure 1a obtained the cogging torque shown in Figure 3 ("original" curve), whose maximum value is 2.32 Nm, while the experimental results of the machine show maximum values of 3.70 Nm. The main reason for this deviation from experimental values is due to component manufacturing tolerance. Consequently, if a tolerance of $\pm0.1$ mm is included in the 20 PMs of the PMSG model and this error is distributed randomly at the height of the PMs, the result shown in Figure 3 (curve "with manufacturing errors") is obtained. Having magnets that are not the same, impacts the cogging torque significantly, mainly because differently sized PMs cause higher magnetic flux variations in the air gap. The maximum cogging torque value obtained in the simulation is 3.90 Nm (Figure 3). This value is slightly higher than the experimental PMSG results, making it possible to validate the developed FEMM model with respect to the cogging torque analysis.

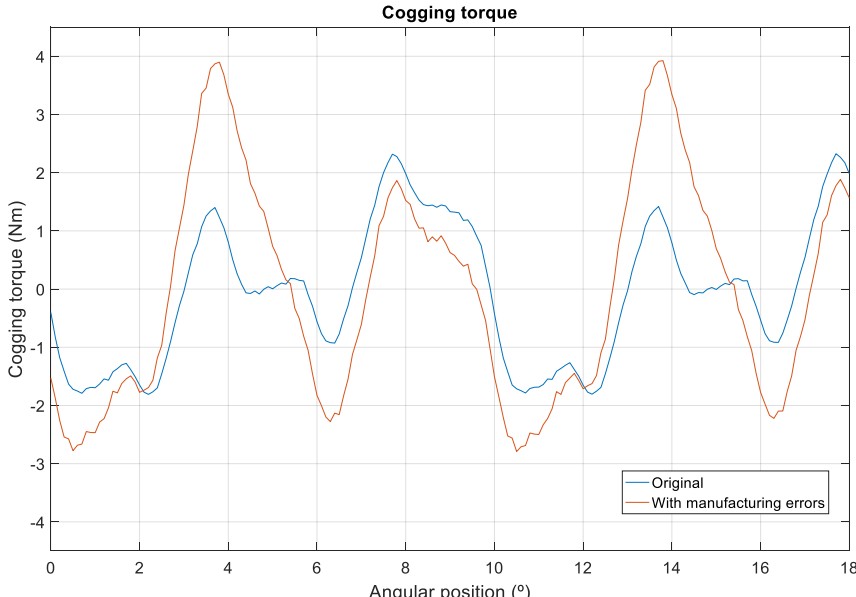

**Figure 3.** Simulation results of cogging torque of the original model considering manufacturing errors.

## 5. Cogging Torque Reduction Methods

### 5.1. Pre-Slot Method

The main objective is to reduce the cogging torque without affecting the machine's construction characteristics and, therefore, without making any changes in the generator's geometry.

A closed-slot stator topology reduces reluctance variation in the air gap and, therefore, the machine's maximum cogging torque value. Furthermore, the minimum dimensions of PMSG slot openings are conditioned by winding mounting factors. Their minimum size depends on the cross section of the winding conductors so that they can be inserted in the slot.

Furthermore, the slot closing method has the drawback of generating a leakage flux through the slots due to the high permeability of the magnetic core connecting the teeth, Figure 4a. These leakages reduce the flux linked by the machine windings, thus producing a drop in induced voltage. This drop in induced voltage can be seen in Figure 5, showing the induced voltage of the PMSG with open slots (the "original" curve) and with closed slots. The effective induced voltage value is 256.4 V in the original generator model with open slots, and it decreases to 221.6 V when the slots are closed.

To avoid the above-mentioned drawbacks, the proposed cogging torque reduction method consists of closing the slots by sliding in a pre-slot part made of the same ferromagnetic material as the stator, as observed in Figure 6. The pre-slot is placed between the teeth longitudinally after machine winding; this does not alter the winding or the slot fill factor. Figure 6a provides details of the space between two of the machine's stator teeth showing where the ferromagnetic part is slid into the start of each

slot. The pre-slot considered in this study is 1.5 mm high; its dimensions are adjusted to the available space to render changing the machine winding unnecessary.

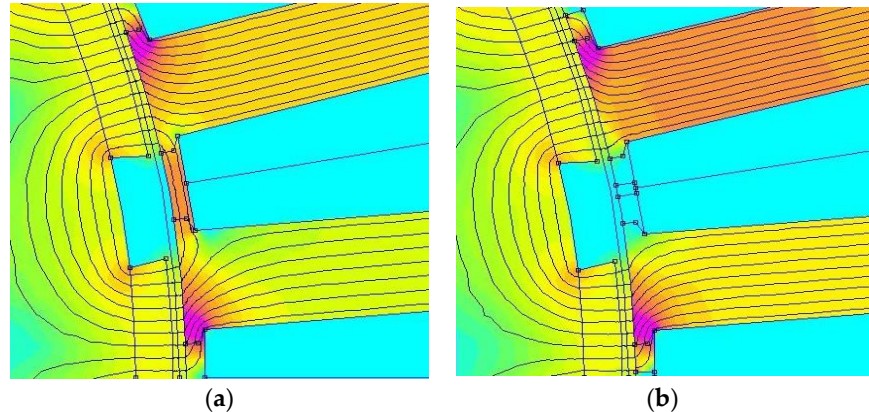

**Figure 4.** Magnetic field lines ($\theta = 0°$): (**a**) Model with closed slots; (**b**) Model with pre-slots.

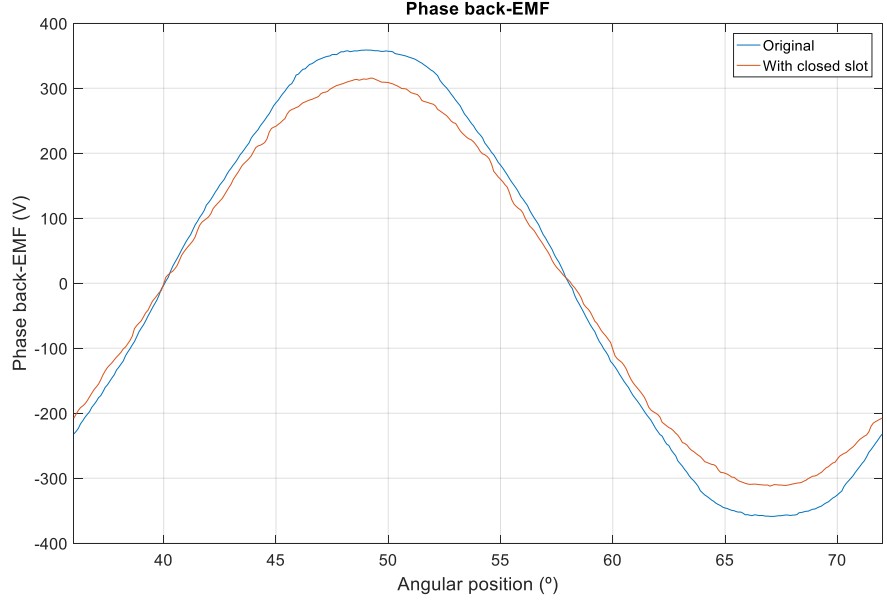

**Figure 5.** Effect of closed slots in back electromotive force (EMF).

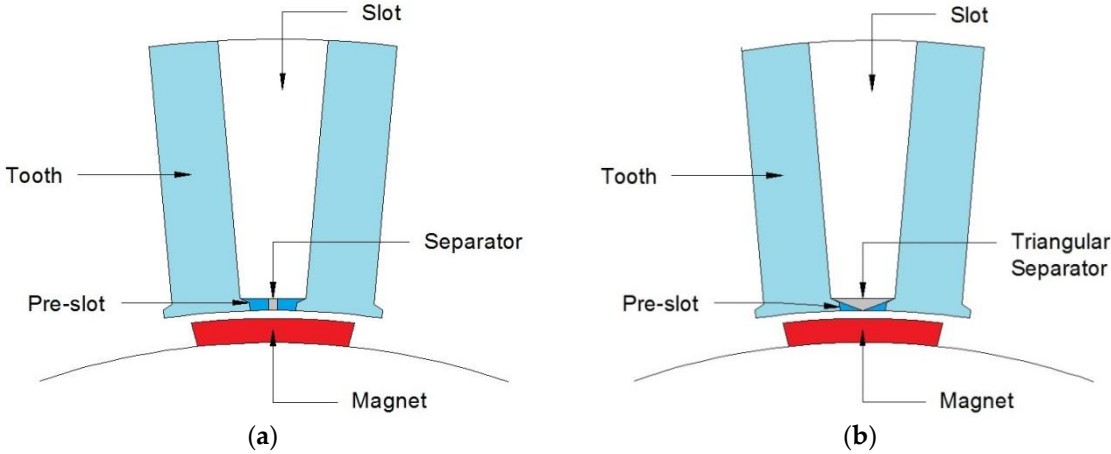

**Figure 6.** Proposed cogging-torque reduction method: (**a**) Pre-slot with separator; (**b**) Pre-slot with triangular separator.

A material separator with low magnetic permeability (aluminum or similar) and a width of 1 mm, the same distance as the machine's air gap, is in the central part of the pre-slot. The purpose of the central separator is to prevent the above-mentioned flux leakage linked by the windings. As this is a non-magnetic separator, it prevents the pre-slot from closing the magnetic field lines and, therefore, preventing flux from circulating between two consecutive PMs.

In accordance with the developed FEMM model, inserting pre-slots with a separator manages to reduce the maximum cogging torque value by 37.9% compared with the original PMSG, as observed in the results shown in the graph in Figure 7, but it does not decrease induced voltage as using the separator reduces leakages.

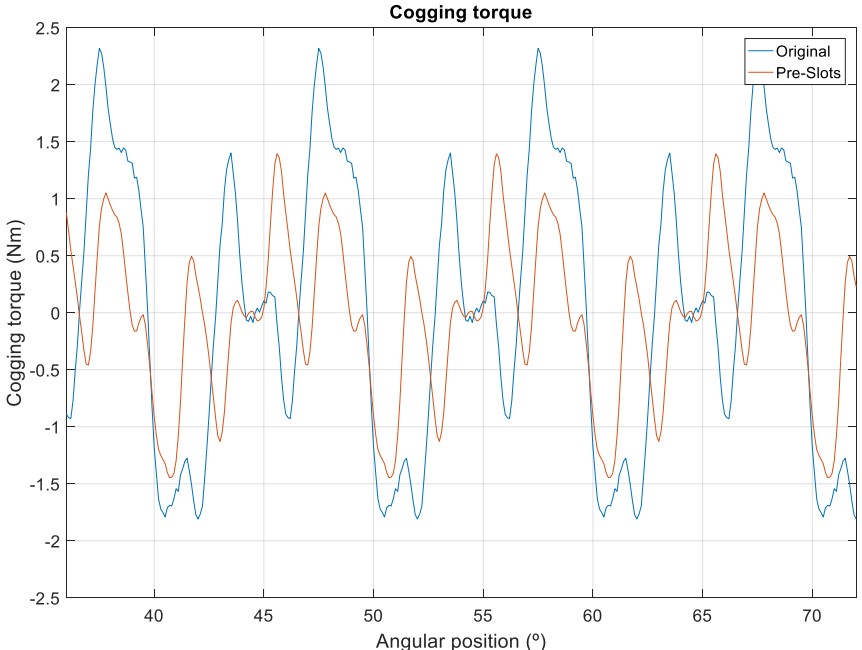

**Figure 7.** Comparison of cogging torque reduction with pre-slot method.

This cogging torque reduction can be improved by considering other alternative geometrical pre-slot configurations. A triangular separator, as shown in Figure 6b, can lessen the magnetic energy variation caused in the original teeth edges or in pre-slot separators, which decreases the machine's cogging torque.

Pre-slot geometry with a triangular separator prevents leakage flux through it, as occurs with the central separator model, thus preventing the undesired decrease in induced voltage and in linked flux through the machine windings. Similarly, it produces a higher reduction in maximum cogging torque, as shown in the graph in Figure 8, decreasing the cogging torque generated by over 47.8%.

An analysis of induced voltage harmonics was conducted to confirm that the installation of the proposed pre-slot system to reduce cogging torque does not affect the machine's technical characteristics. The PMSG is designed for small wind-power applications and, therefore, if an uncontrolled rectifier is used, the harmonics level is of no importance. However, if the connection is via a full converter, the opposite is the case. Figure 9 shows the frequency spectrum of the first 20 harmonics of each of the waves and the harmonic distortion rate (THD) for each model; therefore, the analysis was conducted from the fundamental frequency of 38.67 Hz to the twentieth harmonic of 773.4 Hz.

Table 2 shows how the THD values obtained remain low and similar across all cases and never exceed 2%. The pre-slot solution with a triangular separator presents the least harmonic distortion of the considered models; it is very similar to the closed-slot model and improves the original configuration.

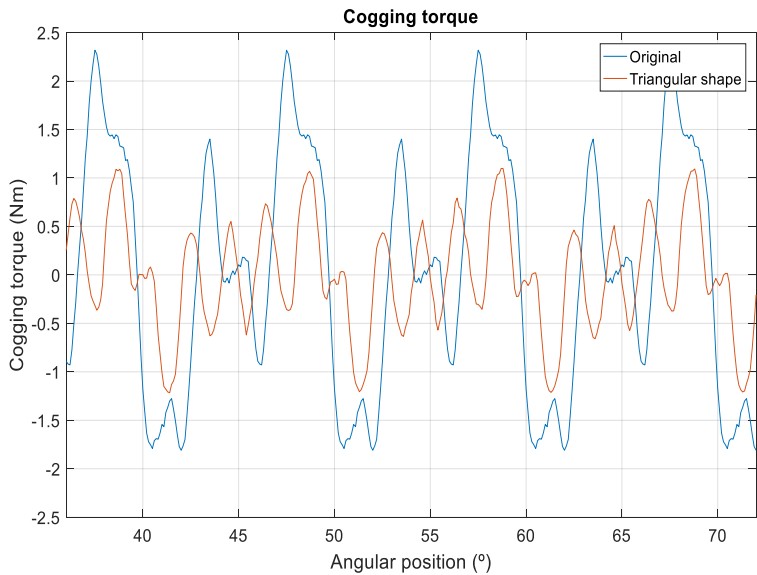

**Figure 8.** Comparison of cogging torque reduction with the triangular pre-slot method.

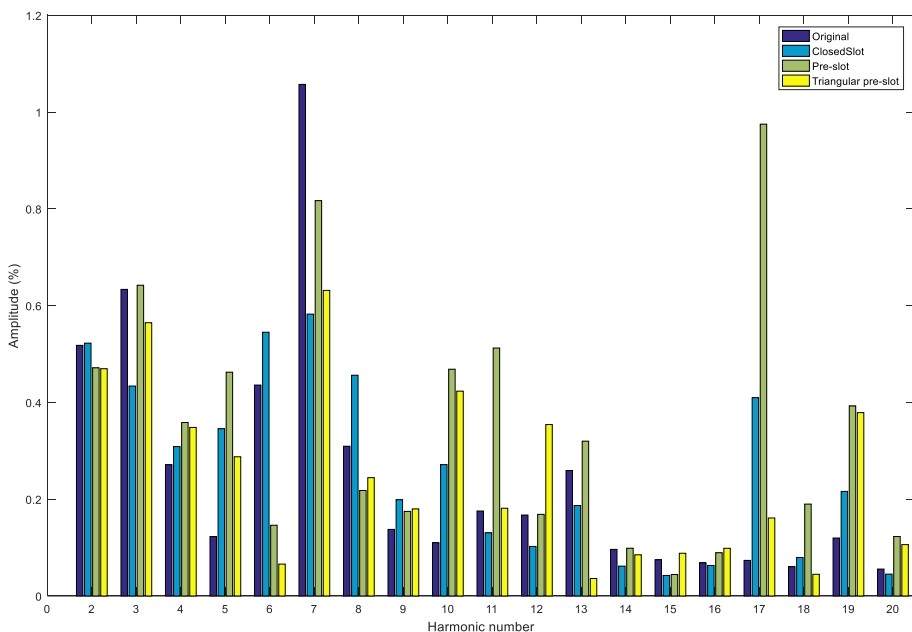

**Figure 9.** Harmonic spectrum for the proposed reduction method.

**Table 2.** Harmonic distortion rate (THD) of the different models.

| Model | THD (%) |
|---|---|
| Original design | 1.54 |
| Design with closed slot | 1.39 |
| Design with pre-slot | 1.88 |
| Design with triangular pre-slot | 1.33 |

### 5.2. Manufacturing Aspects to Reduce the Cogging Torque

Any change in the magnetic circuit alters its reluctance and, therefore, in accordance with Equation (4), it affects the cogging torque and must be considered to reduce it. Consequently, it was found that the holes for correctly aligning the rotor sheet metal with screws in the original design

significantly influence the machine's cogging torque, depending on their position with respect to the PMs, Figure 1b.

The impact of these holes on the cogging torque was analyzed for their different positions with respect to the PMs. The conclusion is that the optimal position, which minimizes cogging torque, is a centered position with respect to the magnets. Figure 10b shows the case in which the rotor hole is centered with respect to the PMs. In this situation, the holes have virtually no influence on magnetic field lines linking one magnet with another. In contrast, when the hole is decentered, Figure 10a, the effect is a smaller effective area in the rotor through which the field lines circulate. Therefore, reluctance increases with respect to the case shown in Figure 10b (centered hole) and the magnetic energy in the rotor decreases as observed in Figure 11.

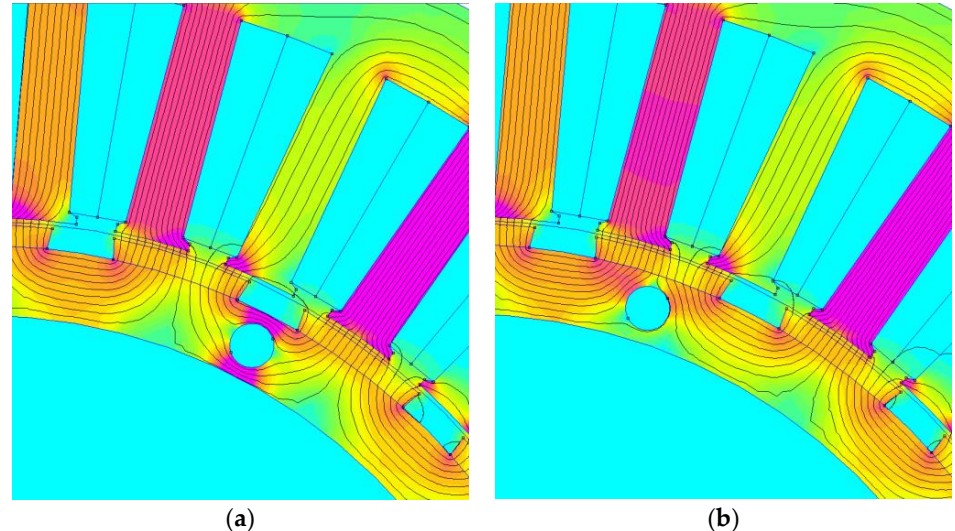

(**a**)　　　　　　　　　　　　　　　(**b**)

**Figure 10.** Magnetic field distribution ($\theta = 0°$): (**a**) Original model, without centered holes; (**b**) Model with centered holes.

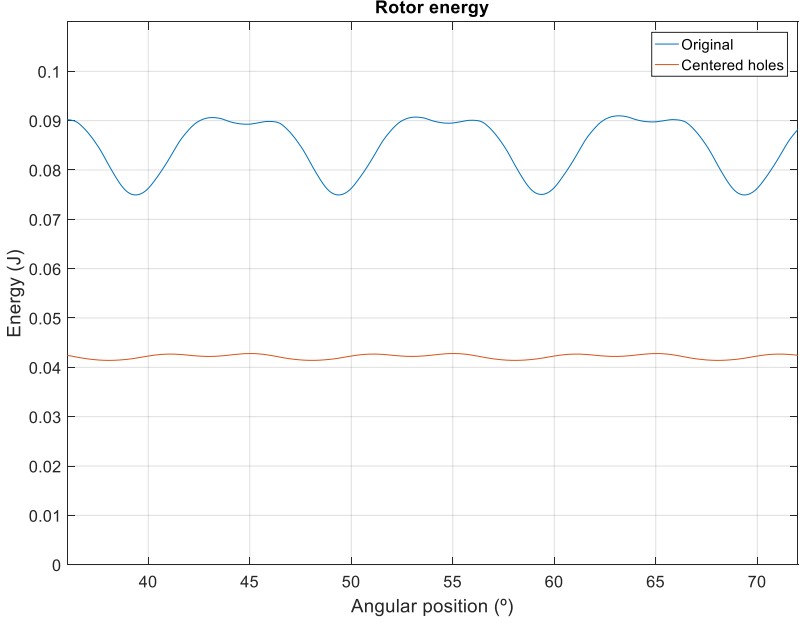

**Figure 11.** Magnetic energy in the rotor.

Figure 12 shows the magnetic energy stored in the machine with respect to the rotor during electrical 360° (mechanical 36°) for both PMSG models. If the hole is decentered, energy minimums

occur when the hole is aligned with a stator tooth. In this position, the flux between two adjacent magnets would be the maximum if there was no hole. Consequently, as observed in Figure 12, and given that the teeth are distributed every mechanical 10° along the stator, the energy minimum occurs with this frequency.

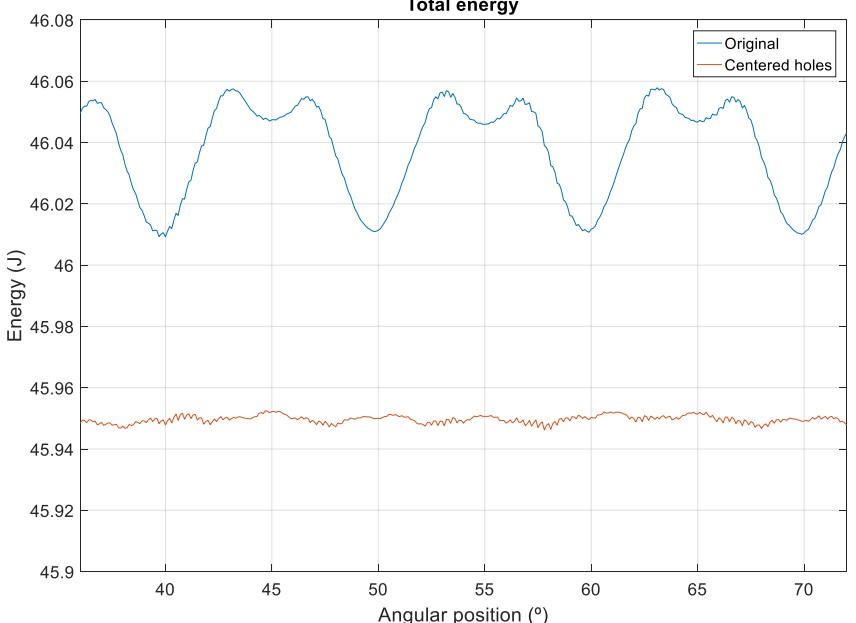

**Figure 12.** Total magnetic energy.

Figure 13 compares the cogging torque for both PMSGs, resulting in a higher value when the hole is decentered given that more magnetic energy variations occur, as presented in Figure 12. The cogging torque of the model with decentered holes is 2.32 Nm, while this value does not exceed 0.86 Nm when the holes are centered. Figure 13 also shows that the presence of decentered holes even produces a change in the cogging torque wave period.

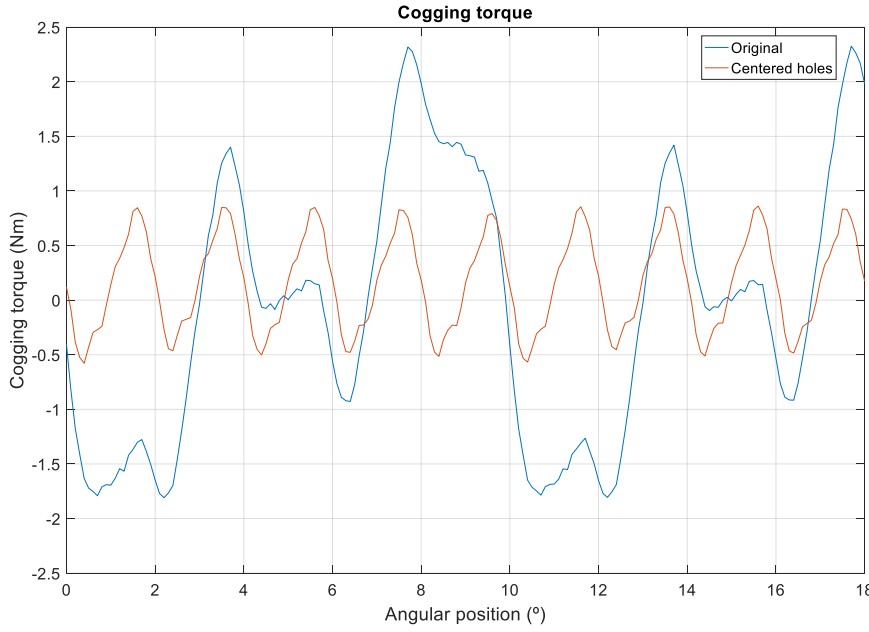

**Figure 13.** Cogging torque of the original PMSG and model with centered holes.

### 5.3. Comparative Results

Table 3 compares the maximum cogging torque values obtained with the original design with the two proposed pre-slot configurations and shows the reduction percentage with respect to the original design. The results of the different models are shown in the ideal case (with no manufacturing errors) and considering manufacturing errors in the magnets. Considering centered holes and the original design, the cogging torque is low (0.86 Nm), however in the real case when considering manufacturing errors this value is still too high in line with experimental measurements (3.31 Nm) with respect to the nominal torque (102 Nm) and needs to be reduced.

**Table 3.** Comparison of the maximum cogging torque values obtained for the prototype and the different models considered in the study.

| Model | Without Manufacturing Errors | | Considering Manufacturing Errors | |
|---|---|---|---|---|
| | Nm | Reduction (%) | Nm | Reduction (%) |
| **Original Design (without Centered Holes)** | | | | |
| Prototype | 3.70 | - | - | - |
| Original design | 2.32 | - | 3.92 | - |
| Pre-slot with separation | 1.44 | 37.9 | 2.03 | 48.2 |
| Triangular pre-slot | 1.21 | 47.8 | 1.90 | 51.5 |
| **Design with all Holes Centered** | | | | |
| Original design | 0.86 | - | 3.31 | - |
| Pre-slot with separation | 0.61 | 29.1 | 1.80 | 45.6 |
| Triangular pre-slot | 0.59 | 31.4 | 1.76 | 46.8 |

Finally, the proposed pre-slot method was compared with the skewing technique and the combination of both is considered. The technique of fractional skewing in the rotor [24] comprises dividing the rotor and turning one division away from another for half the cogging torque period. Four divisions were considered in this analysis, as observed in Figure 14. Therefore, considering that the cogging torque period is mechanical 2°, the shift of one division with respect to another is half of this period, which equals 1°. As observed in Table 4, concerning the model with centered holes and in the ideal case of having no manufacturing errors, applying this combined technique manages to reduce the cogging torque to a peak value of 0.03 Nm or to 0.51 Nm if manufacturing errors are considered. In either of the two cases, the cogging torque reduction is very significant.

**Table 4.** Comparison of the maximum cogging torque values of PMSG with centered holes and the different models considering skewing.

| Model | Without Manufacturing Errors | | Considering Manufacturing Errors | |
|---|---|---|---|---|
| | Nm | Reduction (%) | Nm | Reduction (%) |
| Design with centered holes | 0.86 | - | 3.31 | - |
| Design with centered holes + Triangular pre-slot | 0.59 | 31.4 | 1.76 | 46.8 |
| Design with centered holes + Skewing | 0.31 | 64.0 | 1.34 | 59.5 |
| Design with centered holes + Triangular pre-slot + Skewing | 0.03 | 96.5 | 0.51 | 84.6 |

Figure 15 shows the cogging torque waveform obtained for the PMSG with centered holes. Because of the reluctance periodicity, the cogging torque is a periodic waveform with a frequency given by:

$$f_{T_{cog}} = \frac{\omega \cdot LCM\left(N_{slots}, N_{poles}\right)}{360°} = 696 \text{ Hz} \qquad (6)$$

where $\omega$ is the mechanical speed (1392°/s), $LCM$ the least common multiple of the number of slots ($N_{slots}$ = 36) and the number of poles ($N_{poles}$ = 20). The results in Figure 15 show the decrease in the cogging torque with the pre-slot triangular method and that this improvement is even better when

combining this pre-slot installation technique with fractional skewing in the rotor, up to 84% less in the most realistic case of considering errors in manufacturing processes.

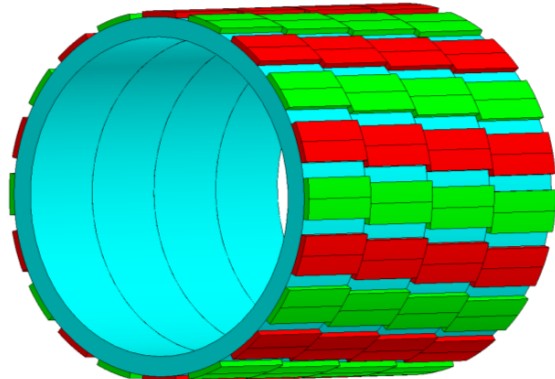

**Figure 14.** Fractional skewing in the rotor.

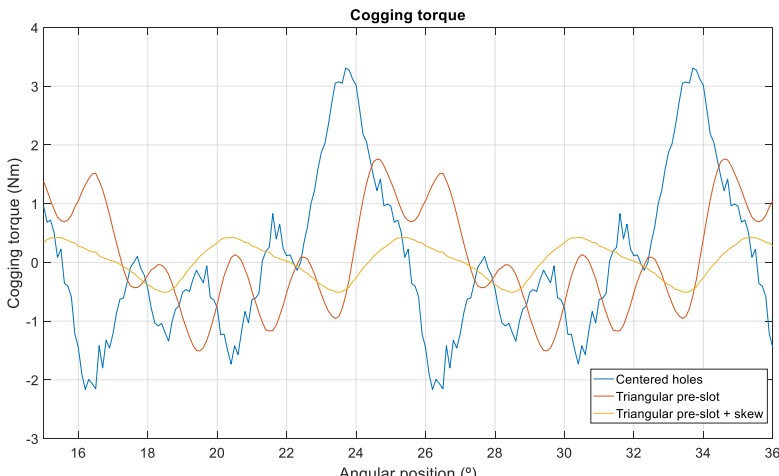

**Figure 15.** Cogging torque of the centered holes PMSG model, triangular pre-slot model and triangular pre-slot model with skewing (with manufacturing errors).

In addition, Figure 16 compares the air gap magnetic flux density in the proposed pre-slot method with the skewing technique, and the combination of both.

Finally, Table 5 shows how the cogging torque reduction affects the machine's induced voltage: the triangular pre-slot method proposed increases EMF due to the magnet flux canalization to the teeth, while skewing decreases it. When combining this pre-slot technique with fractional skewing in the rotor, again a decrease in EMF is observed but in a softer way. As a result, the skew of the rotor reduces the generator efficiency, but combined with the pre-slot technique this reduction will be lower. The cogging torque reduction techniques, as pre-slot technique or skewing, help reducing undesired ripple and allow achieving low cut-in speed in small wind systems; however they add complexity to the manufacturing process and thus may increase the product cost.

**Table 5.** Comparison of the electromotive force (EMF) and cogging torque values of PMSG with centered holes and the different models considering skewing.

| Model | EMF (V) | (p.u.) | Cogging Torque (Nm) |
|---|---|---|---|
| Design with centered holes | 241.52 | 1.000 | 3.31 |
| Design with centered holes + Skewing | 229.47 | 0.950 | 1.34 |
| Design with centered holes + Triangular pre-slot | 247.73 | 1.026 | 1.76 |
| Design with centered holes + Triangular pre-slot + Skewing | 233.04 | 0.965 | 0.51 |

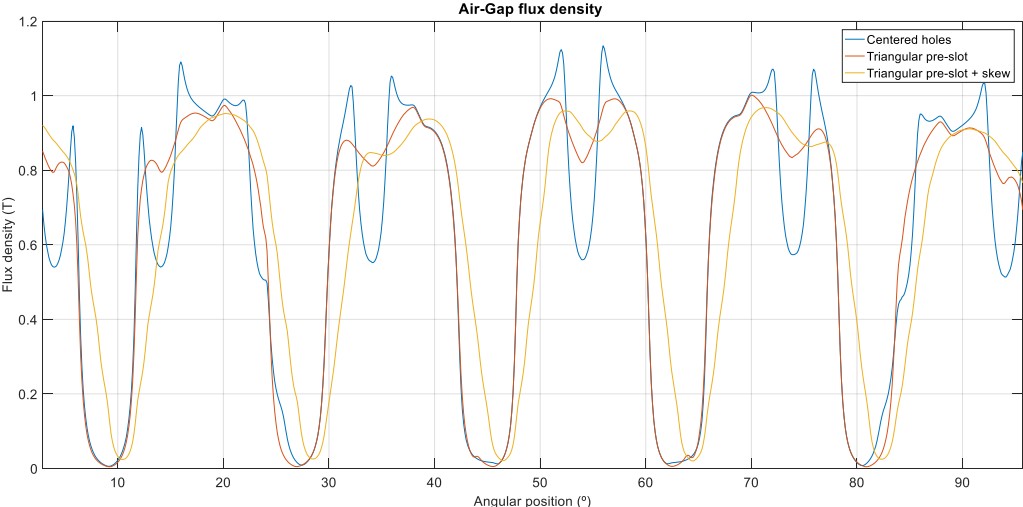

**Figure 16.** Air gap magnetic flux of the centered holes PMSG model, triangular pre-slot model, and triangular pre-slot model with skewing (with manufacturing errors).

## 6. Conclusions

This article presents a new cogging torque reduction technique that does not require changes to the machine's main geometry. It proposes placing pre-slots in the initial part of the stator slots. These pre-slots are made of the same ferromagnetic material as the stator, with a non-magnetic central separator (in two halves). The pre-slots are slid longitudinally between the slots after completing machine winding and, therefore, without altering the PMSG's fill factor.

Introducing a central part of non-magnetic material prevents leakage flux between the machine's teeth and also stops its induced voltage from reducing significantly with respect to the configuration without pre-slots.

The proposed method manages to reduce cogging torque in PMSGs with surface-mounted magnets by up to 47.8%. Additionally, the article analyzes how changing the magnetic circuits for construction reasons can affect the cogging torque, which can easily be optimized. The pre-slot technique is also compatible with other cogging torque reduction techniques, such as skewing. When the above-mentioned methods are combined, cogging torque is reduced by 84.6% considering manufacturing errors.

**Author Contributions:** M.G.-G. and Á.J.R. performed the simulations and wrote the paper. 4fores contributed with the prototype and valuable comments and corrections. All authors discussed the results and commented on the manuscript at all stages.

**Funding:** This research was funded by Spanish Economy, Industry and Competitiveness Ministry and by European Regional Development Fund (ERDF), grant number RTC-2016-5234-3, in the frame of the Project MHiRED "New technologies for wind-photovoltaic hybrid mini networks managed with storage in connection to the grid and with synchronous support in off-grid operation".

**Acknowledgments:** The authors wish to thank 4fores for granting their permission to publish some data presented in this article. Furthermore, the technical support from the Research Group on Renewable Energy Integration of the University of Zaragoza (funded by the Government of Aragon) is also gratefully acknowledged.

**Conflicts of Interest:** The authors declare no conflict of interest.

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
