# Peer review of "Cogging Torque Reduction Based on a New Pre-Slot Technique for a Small Wind Generator"

_energies, doi:10.3390/en11113219_

Round 1
Reviewer 1 Report
The paper presents an interesting method of the cogging torque reduction by the appropriate design of the construction. The final results are almost magnificient (Table 4). However, looking at the problem from the optimization point of view we have two design variables: centering holes (or not) in 2-D cross-sections and the use of fractional skewing the rotor in the third (let say z direction). The objective function is subjected to manufacturing constraints, costs etc. Thus, I can treat the paper as the report of the obtained results without any general remarks dealing with the generality of the proposed methods.
I expect that the authors discuss the possibility of using the proposed methodology in different variants of constructions and expose and express clearly the possible limits of the approach.
The more detailed comparison with other results is required.
Reviewer 2 Report
The paper is better and acceptable now after the authors addressed the reviewers' comments. Thank you so much.
Reviewer 3 Report
The paper presents a intersting method to reduce the cogging torque of SPM generators in wind power applications. With the inter-teeth pre-slot technique, the cogging torque of the example machine could be reduced significantly. However, some comments are suggested in the followings to improve the quality of the paper.
The winding method seems to be dourble-layer concentrated based on shematics given in Fig. 1 but not described in the paragraph. The machine is a 3 phase winding machine listed in Table 1. where the air gap diameter should be rotor diameter or inner diameter of stator.
Equation 5 give the cogging torque calculation scheme without giving mu0 for permiability of air and surface integral with r. However, it seems that most equation is based on rotaional angle phi as the integration variable for the expression.
In Fig. 5, the conventional expression to harmonics is in terms of odd number rather than even number, please verify if there is anly plotting abscissa error?
The cogging torque should not be infinitely reduced to null since the skew of rotor would reduce also the efficiency of the generator. Please make some assessment on this aspect.
Round 2
Reviewer 1 Report
I hope that you will take into account my suggestions and remarks.
Reviewer 3 Report
The paper has been revised with necessary details for readers to comprehend the contents in full scale. Therefore, it is recommeded for immediate publicaiton.